# Reversible single crystal photochemistry and spin state switching in a metal-cyanide complex

Michał Magott [1], Mirosław Arczyński [1,11], Leszek Malec [1], Michał Rams[2], Mathieu Rouzières [3], Andrei Rogalev [4], Fabrice Wilhelm [4], Itziar Oyarzabal [5,6], Thomas Lohmiller [7,12], Alexander Schnegg [8], Coen de Graaf [9,10], Corine Mathonière [3], Rodolphe Clérac [3] & Dawid Pinkowicz [1] ✉

Manipulating the physical properties of solid matter using only photons is a major challenge in materials science. However, achieving such control over a chemical reaction in the solid state is even more challenging. Here we demonstrate the reversible photochemistry occurring in a single crystal of a simple cyanide complex, $K_4[Mo^{III}(CN)_7] \cdot 2H_2O$. Upon exposure to visible light at different wavelengths, a reversible breaking and reformation of dative bonds is triggered, resulting in a photoswitching of the $Mo^{III}$ coordination geometry between 6- and 7-coordinate. This transformation, in turn, induces a spin state change. The observed solid-state photochemical reactivity is robust, quantitative and occurs at a record-high temperature. It paves the way for the development of new photo-switchable high-temperature magnets and nanomagnets.

The absorption of visible light induces substantial changes in the electronic structure of molecules, playing a crucial role in photo-activated processes such as photosynthesis, solar energy harvesting, and photocatalysis. In most cases, the lifetime of the photoexcited state is extremely short. However, certain systems can be thermally trapped in a metastable state for hours, days, or even months before eventually relaxing back to the ground state. This phenomenon enables photoswitching between states with distinct physical and chemical properties. The potential applications of photoswitchable materials are vast, including light-responsive molecular junctions[1], molecular valves and machines[2–4], information storage and processing devices[5,6], and solar energy harvesting[7]. A well-known example of

photoswitching is the photoisomerization of rhodopsin, the protein responsible for vision in humans and many animals, which continues to be an active area of research[8–10].

Many current research efforts in the field of photoswitchable materials focus on light-induced isomerization of organic molecules[11–14] or to a lesser extent – inorganic compounds e.g. transition-metal nitrosyl complexes[15]. While organic photoswitches dominate the field, they do not exhibit changes in magnetic properties upon light absorption with only a few exceptions[16–18]. To achieve the photomagnetic response, they must be coupled with metal centers[19–24]. However, this approach has thus far only produced paramagnetic systems. In contrast, inorganic photomagnetic compounds

¹Faculty of Chemistry, Jagiellonian University, Gronostajowa 2, Kraków, Poland. ²Institute of Physics, Jagiellonian University, Łojasiewicza 11, Kraków, Poland. ³Univ. Bordeaux, CNRS, CRPP, UMR 5031, Pessac, France. ⁴European Synchrotron Radiation Facility (ESRF), 71 av. des Martyrs, Grenoble, France. ⁵BCMaterials, Basque Center for Materials, Applications and Nanostructures, UPV/EHU Science Park, Leioa, Spain. ⁶IKERBASQUE, Basque Foundation for Science, Bilbao, Spain. ⁷EPR4 Energy Joint Lab, Department Spins in Energy Conversion and Quantum Information Science, Helmholtz Zentrum Berlin für Materialien und Energie GmbH, Berlin, Germany. ⁸Max-Planck-Institute for Chemical Energy Conversion, EPR Research Group, Mülheim/Ruhr, Germany. ⁹Departament de Química Física i Inorgànica, Universitat Rovira i Virgili, C. Marcellí Domingo 1, Tarragona, Spain. ¹⁰ICREA, Pg. Lluis Companys 23, Barcelona, Spain. ¹¹Present address: Department of Chemistry, Graduate School of Advanced Science and Engineering, Hiroshima University, Higashihiroshima, Hiroshima, Japan. ¹²Present address: Institut für Chemie, Humboldt-Universität zu Berlin, Berlin, Germany. ✉e-mail: dawid.pinkowicz@uj.edu.pl

typically rely on spin crossover (SCO) sites, which display light-induced excited spin state trapping (LIESST) originating at the metal center[25–28]. Unfortunately, the photoexcited state of most SCO systems suffers from rapid thermal relaxation above 80 K, which limits their practical applications as photomagnetic switches at room temperature. Nevertheless, a careful analysis of the microscopic origins of their photomagnetism offers clues for designing high-temperature magnetic photoswitches. For example, in the $[Fe^{II}(LN_5)(CN)_2]$·MeOH SCO complex, the formation and breaking of a coordination bond extends the lifetimes of its photoexcited state that can be observed up to 130 K[29]. Building on these insights, we propose a strategy towards high-temperature photomagnetic systems based on photochemistry – specifically, the photodissociation of cyanometallates. While cyanide photodissociation has been extensively studied in aqueous solutions[30–33] and, to a lesser extent in aprotic solvents, it has always been found to be irreversible[34]. Photodissociation was also observed in the solid state for $K_4[Mo^{IV}(CN)_8]$·2H_2O[35], but it could only be reversed by thermal treatment.

Herein, we demonstrate a fully reversible metal-cyanide photodissociation occurring in single crystals of potassium heptacyanomolybdate(III) dihydrate[36–38] $K_4[Mo^{III}(CN)_7]$·2H_2O (**1**), and compare it to the photodissociation observed in its solution. The single crystals of **1** undergo a complete metal-cyanide bond photodissociation upon absorption of violet light, resulting in the formation of a hexacoordinate complex, $K_4[Mo^{III}(CN)_6]$·CN·2H_2O (**2**). Remarkably, compound **1** is fully recovered through red light absorption of **2**, representing a solid-state ligand photoassociation reaction. These transformations are non-destructive and maintain the crystallinity of **1** throughout the process. If the unique properties of $[Mo^{III}(CN)_7]^{4-}$ could

be extended to its bimetallic assemblies[35–41], this would enable the light control of the long-range magnetic ordering[39–42], or slow relaxation of the magnetization[43–45] in these compounds.

The photochemical switching behavior of **1** was thoroughly characterized using a combination of techniques, including single-crystal X-ray diffraction (scXRD), infrared (IR) and electron paramagnetic resonance (EPR) spectroscopies, optical reflectivity studies, and X-ray absorption spectroscopy (XAS), complemented by magnetization measurements and X-ray magnetic circular dichroism (XMCD). For comparison, the product of the irreversible photodissociation of **1** in acetonitrile (MeCN) solution, $[K(crypt-222)]_3[Mo^{III}(CN)_6]$·2MeCN (**3**), was also isolated and fully characterized.

## Results and discussion
### Synthesis, solid state photochemistry, and crystallography
**1** was obtained by reacting $MoCl_3(THF)_3$ with KCN in deoxygenated $H_2O$, followed by crystallization from a $H_2O$/MeOH mixture, similar to the method reported by Young[36]. The identity and purity of **1** were confirmed through scXRD, PXRD (Fig. S1), IR spectroscopy (Fig. S2) and elemental analysis (see Methods). The crystal structure of **1** was investigated by using synchrotron radiation (Diamond Light Source, I19) at 30 K. The experiments were conducted on two single crystals, both before and after 405 nm irradiation, followed either by 638 nm irradiation or by heating to 200 K (Fig. 1 and Table S1 with crystallographic details). Before irradiation, the $[Mo^{III}(CN)_7]^{4-}$ anion adopts a capped trigonal prism (CTPR-7) geometry with seven cyanide ligands coordinated to the $Mo^{III}$ center via carbon atoms (Fig. 1a and Table S2). Structural analysis after 20 min of 405 nm irradiation reveals major structural changes in **1**. The most striking feature is the

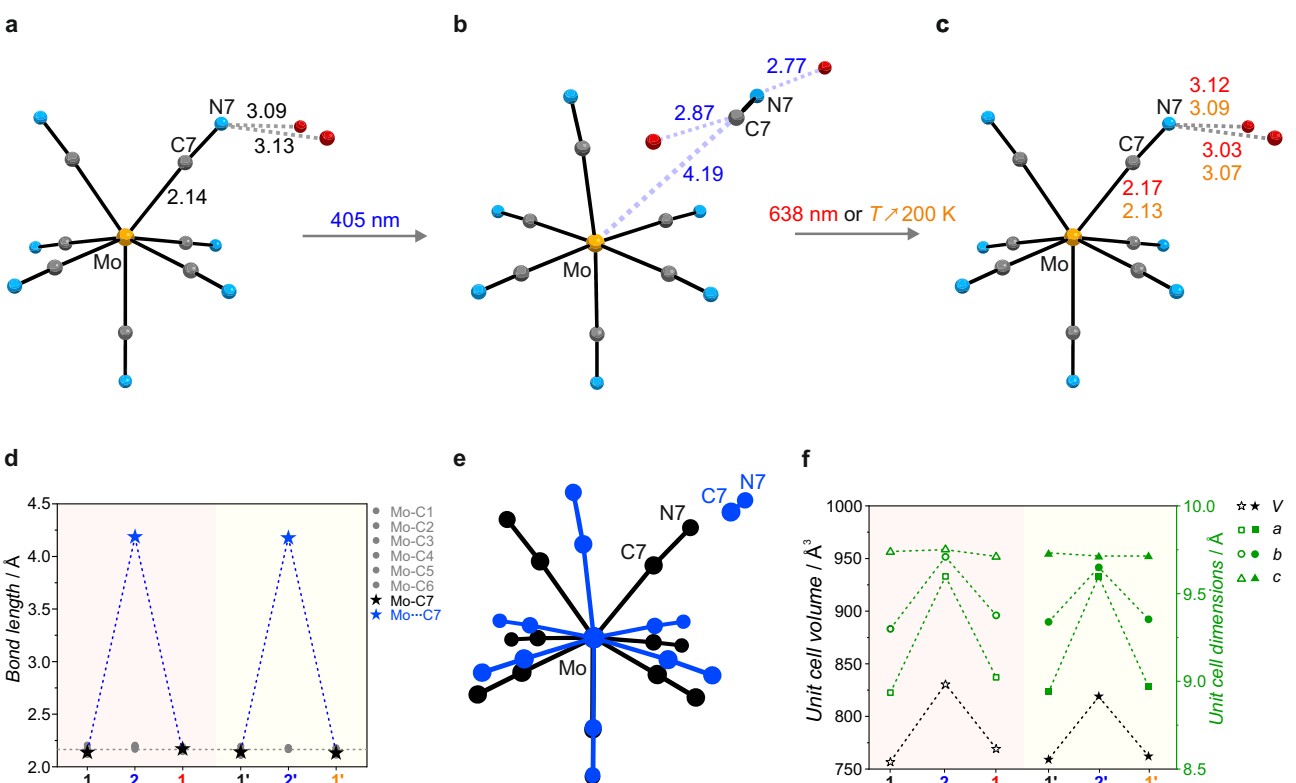

**Fig. 1 | Details of the scXRD photocrystallographic analysis of 1 and its photoproduct, 2, at $T$ = 30 K.** Molecular geometry of the $[Mo^{III}(CN)_7]^{4-}$ anion and selected interatomic distances in Å: **a** before irradiation in **1**; **b** after 405 nm irradiation in **2**; and **c** after 638 nm irradiation (red interatomic distances) or thermal treatment by heating to 200 K (orange interatomic distances); Color code: Mo – yellow, O – red, N – blue, C – gray; Variation of (**d**) the Mo-C bond lengths and (**f**)

unit cell parameters before irradiation (black **1** and **1'**; symbol ' indicates experiments with a second crystal), after 405 nm irradiation (blue **2** and **2'**) and (i) subsequent 638 nm irradiation (red **1**) or (ii) 200-K thermal treatment (orange **1'**). **e** Overlay of the molecular structures of the $[Mo^{III}(CN)_7]^{4-}$ anion in **1** and **2** illustrating the photoinduced changes of the Mo coordination geometry.

disappearance of one cyanide ligand from the Mo coordination sphere. Consequently, the Mo coordination number decreases from seven to six, forming $K_4[Mo^{III}(CN)_6] \cdot CN \cdot 2H_2O$ (**2**; Fig. 1b, e, Supplementary Movie 1). In this photoinduced structure, the remaining six cyanide ligands are arranged at the vertices of a distorted octahedron. The displaced C7N7 cyanide is located at 4.190(9) Å from the Mo center, compared to 2.139(6) Å before irradiation (Fig. 1d), and is trapped between two water molecules and potassium ions, via hydrogen bonds and electrostatic interactions, respectively (Fig. S3). The photodissociation causes all C7N7 cyanide ligands to move in a concerted way into a specific position within the crystal structure of **2** (see Fig. S3 for the packing diagram along the *b* and *a* crystallographic directions for **1** and **2**, respectively). Notably, all other Mo-C bonds in the octahedral complex **2** with the average length of 2.188(11) Å, become more uniform and slightly longer than 2.156(18) Å observed in the pristine **1** (Fig. 1d).

Indeed, the bond lengths in **2** are very similar to those found in $Li_3[Mo^{III}(CN)_6] \cdot 6DMF$ reported by Beauvais and Long[46]. The lattice volume changes from 756.67(17) Å³ for **1** to 830.4(5) Å³ for **2**, representing a 9.7% expansion (black symbols in Fig. 1f and Table S1). This is also reflected in the change of the crystal size upon 405 nm irradiation at 8 K (Supplementary Movie 2). Despite the large displacement of the C7N7 cyanide upon photodissociation to form **2**, the original crystal structure is restored either through 638 nm irradiation or by heating the crystal to 200 K (Fig. 1c and Tables S1, S2). Both restored structures closely resemble the original **1** in terms of molecular features: bond lengths, supramolecular arrangement, and unit cell parameters.

The observed photo-induced changes suggest that the photodissociation of $[Mo^{III}(CN)_7]^{4-}$ leads to the formation of $[Mo^{III}(CN)_6]^{3-}$ and a free $CN^-$ anion trapped between $H_2O$ molecules and $K^+$ cations. This process appears to occur with the retention of the +3 oxidation state of molybdenum in **2**, as indicated by nearly identical Mo-C bond lengths to **3** and $Li_3[Mo^{III}(CN)_6] \cdot 6DMF$ (Fig. S4). The C7-N7 bond length of the photodissociated cyanide in **2** measures 1.19(1) Å, which is typical for a free cyanide, as seen in anhydrous KCN or NaCN[47–49]. This value is only slightly longer than the average C–N bond length observed for cyanide ligands coordinated to Mo in **1** (1.16(1) Å) or in other cyanometallates[50].

## Solution photochemistry

Several transition metal cyanides are known to undergo photolysis in solution[30,32,51]. However, such a behavior was not observed for compound **1**, despite a few reports suggesting that the heptacyanomolybdate(III) anion may transform into hexacyanomolybdate(III)[52–54]. Encouraged by the quantitative and reversible photochemical reactivity of **1** in the solid-state, we decided to test its behavior in a non-aqueous solution. Since **1** is insoluble in organic solvents, [2.2.2] cryptand (crypt-222)[55] was added to dissolve it in anhydrous acetonitrile. The resulting yellow solution was irradiated with either violet or white light (Fig. S5), and in both cases, bleaching occurred within minutes (Fig. S6). This is consistent with the photobleaching of a single crystal of **1** observed at 8 K (Supplementary Movie 2). Colorless plate crystals of $[K(crypt–222)]_3[Mo^{III}(CN)_6] \cdot 2CH_3CN$ (**3**) were isolated as the sole Mo-based product from this solution (Table S3 and Figs. S7, S8) while the by-product [K(crypt-222)]CN remains in solution. The isolated compound showed no electronic transitions in the visible range of the UV-vis spectrum, either in the solid state or in MeCN solution (Fig. S9), supporting that bleaching results from the light-induced dissociation of a metal-cyanide bond. The scXRD analysis of **3** revealed a nearly identical geometry of the $[Mo^{III}(CN)_6]^{3-}$ anion to that observed in **2** (see Supplementary Information for detailed discussion). Thus, the photochemical reactivity of $[Mo^{III}(CN)_7]^{4-}$ in aprotic solution mirrors the solid state behavior, though it is irreversible in solution.

## Spectroscopic characterization

Optical reflectivity measurements were conducted on **1** in the solid state, revealing broad bands that are nearly temperature independent (Fig. S10a, b). Exposure to specific wavelengths within the 365–1050 nm range at 10 K produced an increase in reflectivity below 800 nm, particularly under 365, 385 and 405 nm irradiations (Fig. S10c–f), which again is in line with the bleaching of the green crystals of **1** upon 405 nm irradiation producing nearly colorless **2** (Supplementary Movie 2).

To obtain the spectrum of the photoexcited state, which persists up to 150 K (Fig. S10f), **1** was irradiated for 2 h at 405 nm—the same wavelength used in photocrystallography (Fig. S10e). Above 150 K, the characteristic bands of the photoexcited state begin to fade, disappearing entirely above 160 K as the original spectrum of **1** is restored (Fig. S10e, f). Additionally, **1** and its photo-induced state **2** can be reversibly cycled using 385 and 660 nm irradiations over multiple cycles (Fig. S11).

The reversible photodissociation of the metal-cyanide bond was also confirmed by IR spectroscopy (Fig. 2a) in a polycrystalline sample of **1**. The $C \equiv N$ stretching vibrations give strong IR absorption bands, highly sensitive to the metal center's electronic configuration and the complex's geometry and symmetry. At 100 K, the $\nu(C \equiv N)$ band for **1** appears as two peaks: a main one at 2069 cm⁻¹ and a weaker component at 2105 cm⁻¹, consistent with the literature values[37]. Following 405 nm irradiation, the band maximum shifts to 2104 cm⁻¹ with a shoulder at 2073 cm⁻¹. The main peak position in **2** aligns well with the cyanide stretching observed in $Li_3[Mo^{III}(CN)_6] \cdot 6DMF$, reported at 2115 cm⁻¹ [46], with a small difference highlighting the influence of the second coordination sphere of $[Mo^{III}(CN)_6]^{3-}$. Further support comes from the IR spectrum of **3**, which shows a single, narrow $\nu(C \equiv N)$ band at 2088 cm⁻¹ (Fig. S12). It is worth noting that heating **2** to 200 K fully restores the initial spectrum typical for **1** (Fig. 2a).

XAS measurements at 4 K before and after irradiation support also the photo-induced formation of $[Mo^{III}(CN)_6]^{3-}$ (Fig. 2b). In the initial Mo $L_{2,3}$-edge spectra of **1**, prominent white line resonances arise from $2p \rightarrow 4d$ excitations, with fine structures reflecting ligand field splitting of the $4d$ states. Following 405 nm irradiation, the two primary signals weaken, while lower-energy peaks intensify. This aligns with the capped trigonal prism geometry of the $[Mo^{III}(CN)_7]^{4-}$ anion, characterized by a single electron hole in the $[(4 d_{x2-y2})(4 d_{xz})]^3$ orbital pair[56], leading to a lower intensity of the $2p \rightarrow [(4d_{x2-y2})(4d_{xz})]$ transition (Fig. 2c). In contrast, the nearly octahedral geometry of the photo-induced $[Mo^{III}(CN)_6]^{3-}$ corresponds to the $t_{2g}^3 e_g^0$ configuration with three holes in the $t_{2g}$ orbitals. This results in a higher intensity of the low energy pre-edge region at the expense of the main resonances, mirroring the comparison between the XAS spectra of **1** and **3** (Fig. S13). The integrated white-line intensities for **1**, **2**, and **3** are nearly identical, indicating consistent $4d$ occupancy across these compounds. Further insights into the unoccupied Mo $4d$ orbitals were provided by the XMCD measurements, showing that the dichroic signals for **2** are predominantly concentrated at the low energy peaks for both Mo $L_2$ and $L_3$ edges (light blue line in Fig. 2b).

Application of the magneto-optical sum rules[57,58] to the XMCD spectra reveals contributions from both spin magnetic moment ($M_S$) and orbital magnetic moment ($M_L$) in the magnetization of the photoproduct **2**. The calculated values of $M_S = 2.86\ \mu_B$ and $M_L = -0.16\ \mu_B$ agree well with those expected for high-spin hexacyanomolybdate(III), where the $t_{2g}^3 e_g^0$ electron configuration implies a minimal orbital contribution.

The solid-state EPR spectrum of powdered **1** is consistent with the findings of Rossman et al.[37] showing an $S = \frac{1}{2}$ system with slight *g*-factor anisotropy ($g_z = 2.108$, $g_{xy} = 1.964$, black line in Fig. 2d). Upon 405-nm irradiation at 10 K, the $S = \frac{1}{2}$ signal gradually decreases and vanishes within tens of minutes, while a new broad resonance appears around 160 mT, corresponding to $g' \approx 4.3$ (blue line in Fig. 2d). This EPR

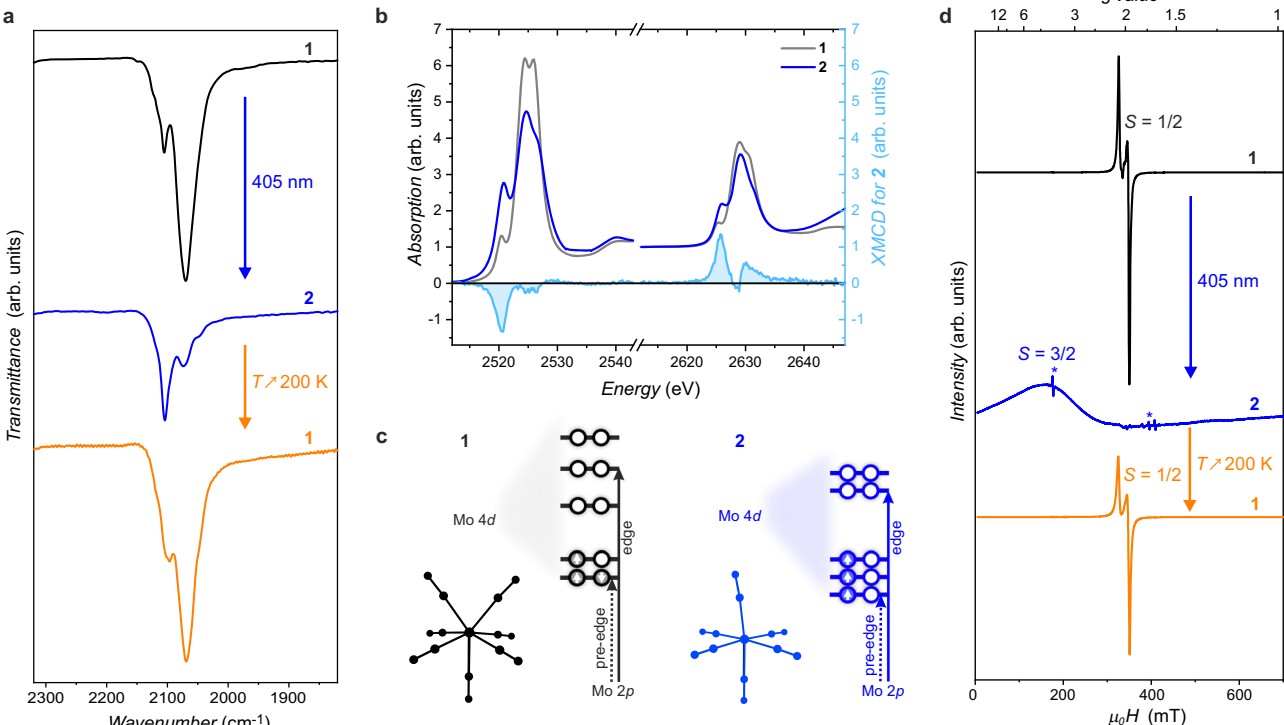

**Fig. 2 | Spectroscopic studies (IR, XAS/XMCD and EPR) of 1 and its photo-product, 2. a** Comparison of the IR spectra shown in the 1820–2320 cm⁻¹ range and recorded at 100 K before irradiation (**1**, black line), after 405-nm irradiation (**2**; blue line), and after thermal relaxation at 200 K (**1**; orange line). **b** XAS spectra recorded at 4 K for **1** (before irradiation; gray line) and for **2** (after 405-nm irradiation; dark blue line), and XMCD signal for **2** (light blue line with blue filling). **c** Scheme illustrating the change of the XAS intensity of the respective Mo $2p \rightarrow 4d$ transitions (at the pre-edge and edge) related to the electronic configurations of **1** (capped trigonal prism geometry) and **2** (pseudo-octahedral geometry). **d** X-band EPR spectra recorded at 10 K before irradiation (**1**, black line), after 405 nm irradiation (**2**, blue line; multiplied by a factor of 20), and after thermal relaxation (**1**, orange line). Asterisks mark the resonator background signals in the spectrum of **2**.

signal is attributed to the anticipated $S = 3/2$ state of **2**, as inferred from the XMCD study. Following thermal relaxation at room temperature, the initial $S = ½$ spectrum of **1** is recovered, while the 160 mT signal vanishes (orange line in Fig. 2d), further confirming the reversible transformation observed in crystallographic, optical reflectivity, and IR studies.

### Computational studies

To investigate the photoswitching mechanism, ab initio calculations (CASSCF/CASPT2) of the electronic transitions in **1** and **2** were conducted. For **1**, the light-driven transformation to **2** likely involves either a spin-allowed transition calculated at 361.5 nm or spin-forbidden ones at 373.4 and 460.9 nm (Table S5). However, in the solid state, the absorption bands are broad and heavily overlap, allowing efficient photoconversion to **2** experimentally at 365, 385, and 405 nm, as observed in optical reflectivity studies (Fig. S10c).

In contrast, the calculated spin-allowed transitions for **2** fall within the deep UV region, while the $^4A_{2g} \rightarrow \, ^2T_{2g}$ transitions appear at 587.6 and 604.8 nm, matching the de-excitation wavelengths (Table S6 and Fig. S11a). These spin-forbidden transitions gain intensity due to a slight distortion in the octahedral geometry of $[Mo^{III}(CN)_6]^{3-}$ observed in **2**, potentially enabling the light-induced reverse metal-cyanide bond association process.

Periodic DFT geometry optimizations accurately reproduce changes in the metric parameters of the crystal structures and the molecular geometry of the complex associated with a spin-state change at the Mo^III center. The **1 → 2** transformation was reproduced by re-optimizing the low-spin state structure of **1**, fixing the number of unpaired electrons per Mo to 3 within the first optimization step, as inferred for the high-spin state structure of **2** from EPR and XMCD (see Methods and Supplementary Information for details). The reverse **2 → 1** transformation was similarly modeled by re-optimizing the structure of **2** for one unpaired electron per Mo. Figs. S14–S17 and Tables S7–S8 compare the experimental and optimized geometries of the $[Mo^{III}(CN)_7]^{4-}$ and $[Mo^{III}(CN)_6]^{3-}$ anions in their respective crystal structures.

### Photoswitching of the magnetization

The magnetic and in situ photomagnetic studies of **1** were conducted using variable temperature/magnetic field magnetometry with light delivered to the sample chamber through an optical fiber. The temperature dependence of the $\chi T$ product (product of the molar magnetic susceptibility, $\chi$, and temperature, $T$) measured at 0.1 T for pristine **1** shows behavior typical of an $S = ½$, $g = 2.0$ paramagnetic system (Fig. S18a), in agreement with the EPR results (Fig. 2d). Upon cooling below 50 K, the $\chi T(T)$ data deviates from the constant value of 0.36 cm³·K·mol⁻¹, ultimately decreasing to 0.03 cm³·K·mol⁻¹ at 1.8 K, indicating supramolecular antiferromagnetic interactions between $S = ½$ Mo^III spins. Below 3.5 K, these couplings lead to an antiferromagnetically ordered state, as evidenced by the $M(H)$ data ($M$ - molar magnetization, $H$ - magnetic field strength; Fig. S18b) and corroborated by low-temperature heat capacity measurements (Fig. 19b), discussed in detail in the Supplementary Information (accompanied by Figs. S18–S20).

Photomagnetic experiments (Fig. 3) conducted on **1** at 10 K show an increase in $\chi T$ from 0.30 to 1.79 cm³·K·mol⁻¹ upon 405-nm irradiation (ca. 5 mW·cm⁻², 2 h, violet and black symbols in Fig. 3a). The $\chi T(T)$ profile recorded in the dark, immediately after irradiation, reveals a plateau around 1.84 cm³·K·mol⁻¹ in the 20-100 K range (blue symbols in Fig. 3b) close to 1.875 cm³·K·mol⁻¹ expected for a magnetically isotropic

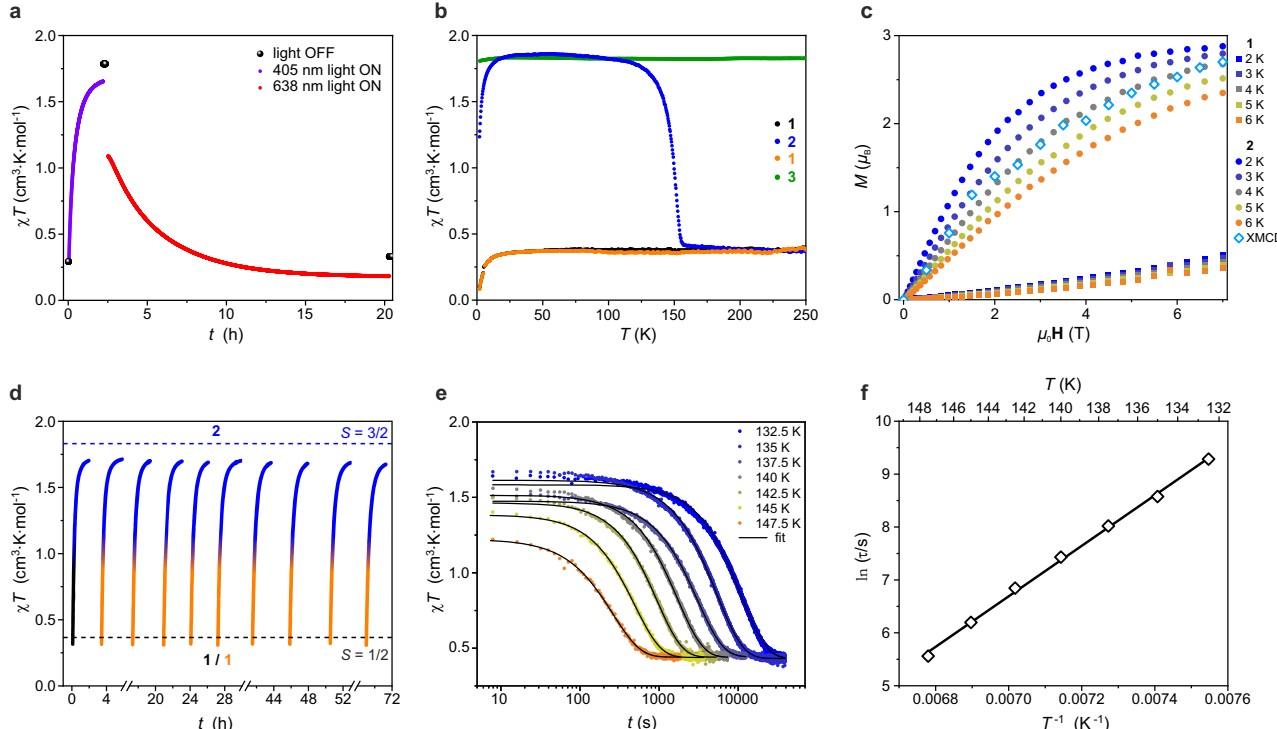

**Fig. 3 | Magnetic and photomagnetic studies of 1. a** Time evolution of the $\chi T$ product of **1** at 0.1 T during 405-nm irradiation (purple symbols), converting **1** into **2**, and subsequent 638-nm irradiation (red symbols), restoring **1** from **2** ($T = 10$ K; note that no correction was applied for the thermal heating due to light irradiation, with black symbols representing the $\chi T$ product in the absence of light). **b** Temperature dependence of the molar magnetic susceptibility (shown as the $\chi T$ product; at 0.1 T) for **1** (before irradiation, black points), **2** (after 405-nm irradiation, blue points) and **1** (after thermal relaxation of **2**, orange points). For comparison, the green points show $\chi T(T)$ for the reference $[Mo^{III}(CN)_6]^{3-}$ product **3**. All curves were recorded at a 2 K·min$^{-1}$ heating rate. **c** Magnetization ($M$) versus magnetic field ($H$) plots for **1** (before irradiation, square symbols) and **2** (after 405 nm irradiation, circle symbols) recorded between 2.0 and 6.0 K, compared to XMCD signal

(diamond symbol) at the Mo $L_3$ edge at 2520 eV for **2** at 4 K. **d** Time evolution of the $\chi T$ product (at 0.1 T) of **1** during 10 consecutive cycles of 405-nm irradiation at 10 K, each followed by thermal relaxation at 200 K, confirming reversible photo-induced transformation at the bulk level. **e** Time evolution of the $\chi T$ product (at 0.1 T) recorded for **2** in the dark after 405-nm irradiation of **1** at 10 K and subsequent heating to target temperatures in the 132.5-147.5 K range (heating rate 15 K·min$^{-1}$). Each relaxation study was followed by heating to 200 K to ensure complete restoration of **1**. Solid lines represent fits to stretched exponential decay functions, providing relaxation times, $\tau$. **f** Plot of the relaxation time, $\tau$, as a function of $T^{-1}$ (diamond symbol), with an Arrhenius fit (solid line) depicting the thermally activated relaxation process of **2** reverting to **1**.

$S = 3/2$ state (with $g = 2$). This aligns with the 1.82 cm³·K·mol$^{-1}$ value observed for **3** (green symbols in Fig. 3b; for details see Supplementary Information and Figs. S21-22). Additionally, the $M(H)$ dependence for **2** shows a magnetization close to 3.0 $\mu_B$ at 7 T, indicating a high-spin $S = 3/2$ configuration (Fig. 3c), in good agreement with XMCD results and the magnetic data for **3** (Fig. S22). Collectively, these photomagnetic experiments at 10 K suggest thermal trapping of the $S = 3/2$ metastable state, analogous to spin-crossover systems, confirming the formation of light-induced high-spin hexacyanomolybdate(III) complex. As the temperature rises (at 2 K·min$^{-1}$), the photoinduced state relaxes to the original $S = \frac{1}{2}$ ground state of the $[Mo^{III}(CN)_7]^{4-}$ complex around 150 K, indicated by a sharp drop in the $\chi T$ product (blue symbol in Fig. 3b). Notably, the relaxation temperature of 150 K for **2** surpasses that of previously reported spin-crossover systems[19,20,59], suggesting an unusually long lifetime of the photoinduced state in this molybdenum cyanide complex. To quantify the characteristic time of the thermally induced **2** → **1** relaxation, time-dependent magnetization studies were performed at several temperatures in the 132.5-147.5 K range (Fig. 3e). The temperature dependence of the estimated relaxation time (Table S9) follows an Arrhenius behavior with an activation energy of 4770(90) K (Fig. 3f). Hence, with the increasing temperature, high-energy states become populated enabling **2** to overcome the energy barrier of cyanide re-association and relaxation to **1**. The relaxation behavior of **2** is comparable with that reported for the record-holding photomagnetic molecular Prussian blue analog $[Co_4Fe_4]^{60}$. This result

indicates that considerable thermal energy is needed for the complex to overcome the energetic barrier between the high-spin and low-spin states. This aligns with the substantial structural reconfiguration upon the **2** → **1** relaxation − specifically, the displacement of the C7N7 cyanide group by about 2 Å back towards the Mo center. As shown in Fig. 3d, the robustness and reversibility of the 405 nm irradiation at 10 K (**1** → **2**) and thermal relaxation (**2** → **1**) processes was confirmed across multiple cycles by magnetization measurements. Moreover, similar behavior is observed at the record-high temperature of 100 K over at least four consecutive cycles of 405 nm irradiation and thermal relaxation (Fig. S23) with only slight radiation damage to the sample.

Importantly, the substantial energy barrier of the thermally activated metal-cyanide association (**2** → **1**) does not prevent this process from happening under 638 nm irradiation at low temperatures as demonstrated by scXRD at 30 K (Fig. 1) or optical reflectivity at 10 K (Fig. S10). Photomagnetic experiments at 10 K further support this observation (Fig. 3a): following the initial 405 nm irradiation of **1** at 10 K, which photoconverts **1** into **2**, a subsequent 638 nm irradiation fully restores the magnetic susceptibility of **1** (red symbols in Fig. 3a).

To summarize, a single crystal of the inorganic compound $K_4[Mo^{III}(CN)_7]\cdot 2H_2O$ demonstrates a reversible photochemical reaction, enabling the molybdenum center to switch between the 7-coordinate low-spin and 6-coordinate high-spin geometries. Exposure to 405 nm light causes dissociation of one of the seven metal-cyanide bonds, resulting in a 2 Å displacement from the Mo center,

while subsequent irradiation at 638 nm reverses this effect, restoring the original structure. This bidirectional photoswitching is remarkably robust and non-destructive, yielding colossal structural, electronic, and magnetic changes. Interestingly, photodissociation in solution is irreversible, permitting isolation of the $[Mo^{III}(CN)_6]^{3-}$ complex.

$K_4[Mo^{III}(CN)_7]\cdot 2H_2O$ stands out as a unique example of bidirectional photoswitching in an inorganic coordination compound in the crystalline phase. This contrasts with the previously reported $K_4[Mo^{IV}(CN)_8]\cdot 2H_2O$[35] where cyanide re-association proceeds exclusively through the thermal route. The high performance of $K_4[Mo^{III}(CN)_7]\cdot 2H_2O$ is highlighted by three key factors: (i) photo-induced ligand dissociation achieves 100% efficiency, (ii) the photo-dissociation is fully reversible with light of a different wavelength resulting in photo-association, and (iii) the record-high relaxation temperature allows photoswitching at temperatures exceeding 100 K. These exceptional photochemical properties set the stage for designing new photomagnetic materials that could switch between paramagnetic and ferromagnetic (or ferrimagnetic) states at room temperature. By leveraging similar mechanisms in coordination polymers and materials based on the $[Mo^{III}(CN)_7]^{4-}$ building-block, or its analogs such as the aforementioned $[Mo^{IV}(CN)_8]^{4-}$, this approach holds promise for advanced high-temperature photomagnetic applications.

## Methods

### Syntheses

All reagents other than the ones described below were purchased from commercial sources and used without further purification. The distilled water used in the synthesis of **1** was deoxygenated by refluxing demineralized water under an argon atmosphere for at least 12 h. Deoxygenated methanol, acetonitrile, and diethyl ether were prepared by passing HPLC grade solvents through the Inert PureSolv EN7 solvent purification system.

**MoCl₃(THF)₃** precursor is commercially available, but can also be synthesized under argon atmosphere inside a glovebox in a direct two-step reduction of $MoCl_5$ according to a modified literature method[61]. 6.0 g of $MoCl_5$, 15.0 g of Sn (coarse powder 20 mesh or 5 mm shots) and 70 ml of $Et_2O$ were placed in a flask and stirred for 3 h. The supernatant liquid was decanted from a highly crystalline brown solid and replaced with 50 ml of THF. The suspension was stirred for another 30 min (for Sn powder) or 3 h (for Sn shots). The reaction mixture changed color from very dark brown-orange to orange (orange precipitate plus purple solution). The suspension was separated from excess tin by decantation and then vacuum filtered using a P3 glass fritted funnel. The crystalline solid was washed with THF (2 × 10 ml), followed by vacuum drying (15 min), which yielded a salmon-orange crystalline solid. Yield: 3.4 g (ca. 37%). The purity and identity of the product as $MoCl_3(THF)_3$ was confirmed by EA and PXRD.

**K₄[Moᴵᴵᴵ(CN)₇]·2H₂O (1)** was synthesized using a modified procedure based on established literature methods[36,62]. Inside the oxygen-free glovebox ($O_2 < 0.1\%$), a Schlenk flask was charged with $MoCl_3(THF)_3$ (1.7 g, 4.1 mmol), KCN (3.0 g, 46.0 mmol; Sigma-Aldrich, ≥96%) and 20 mL of deoxygenated water. The flask was sealed with a glass stopper, removed from the glovebox and connected to the Schlenk line. The mixture was subjected to vigorous magnetic stirring, evacuated slowly evacuated until it began to boil gently and then sealed. The flask was wrapped in an aluminum foil to prevent light exposure and heated to 50 °C for 24 h. After that the reaction mixture was transferred to an oxygen-free glovebox and vacuum-filtered through a sintered-glass funnel (G4) to remove any insoluble impurities. The filtrate was treated with 13 mL of deoxygenated methanol (MeOH), sealed in the glass jar and kept at 4 °C for more than 24 h. After that time, the precipitation of dark-green crystals was observed, which were collected by vacuum filtration inside oxygen-free glovebox and washed with MeOH:H₂O = 7:3, MeOH:H₂O = 9:1 and pure MeOH, then dried in vacuo for 15 min. The final product (1.5 g) was obtained in

78% yield, based on $MoCl_3(THF)_3$. Anal. Calcd. (found) for $C_7H_4K_4MoN_7O_2$, C: 17.87% (18.01%), H: 0.857% (0.873%), N: 20.84% (20.94%). The thermogravimetric analysis indicates the complete loss of water molecules of crystallization at 80 °C in a single step (Fig. S24).

**[K(crypt−222)]₃[Moᴵᴵᴵ(CN)₆]·2MeCN (3)** was synthesized via a photochemical route, similar to that utilized in photochemical preparation of $[K(crypt-222)]_3[Mo^{IV}(CN)_7]\cdot 3MeCN$ and $[K(crypt-222)]_3[W^{IV}(CN)_7]\cdot 4MeCN$[35,63]. In the Inert PureLab HE glovebox ($O_2 < 0.1$ ppm and $H_2O < 0.5$ ppm), $K_4[Mo^{III}(CN)_7]\cdot 2H_2O$ (0.24 g, 0.51 mmol) was placed in a vial filled with 7 mL of deoxygenated $CH_3CN$ in the presence of [2.2.2]cryptand (crypt-222; 0.9 g, 2.39 mmol; Merck, ≥99%). The yellow suspension was irradiated using white light (Photonic LED-Light-Source F3000) until it turned into a nearly colorless solution. Then it was subjected to slow diffusion of diethyl ether vapor for two days, resulting in the precipitation of colorless plate crystals, which were collected by vacuum filtration. The purity of the product varies from batch to batch; at elevated ambient temperatures, increased evaporation of diethyl ether can lead to the precipitation of a by-product, [K(crypt−222)]CN. In this case, the product can be purified by dissolving it in 4 mL of $CH_3CN$ and then performing diffusion of diethyl ether vapors. The final product (0.7 g) is obtained in 87% yield based on $K_4[Mo^{III}(CN)_7]\cdot 2H_2O$. The purity of the compound was checked by PXRD, with the experimental pattern (Fig. S7) matching perfectly the simulated one from the scXRD structural model obtained at 293 K (CCDC 2352266).

**Structure determination and refinement.** Single-crystal X-ray diffraction experiments for the 1ˢᵗ crystal of pristine **1** (CCDC 2352257), after 405 nm irradiation **2** (CCDC 2352263), after 638 nm irradiation **1** (2352259), for the 2ⁿᵈ crystal of pristine **1′** (CCDC 2352264), after 405 nm irradiation **2′** (CCDC 2352261), after thermal relaxation at 200 K **1′** (CCDC 2352262) and for two test crystals **1test** (CCDC 2352258) and **1test2** (CCDC 2352260) were all performed at 30 K using synchrotron X-ray radiation (Diamond Light Source, United Kingdom) at the I19 beamline (EH2) equipped with Newport 4-circle κ-diffractometer, monochromatic X-ray synchrotron radiation (λ = 0.6889 Å), Pilatus 300 K detector and an N-Helix Oxford Cryosystems gas cooler[64]. In situ irradiation experiments were conducted with 405 nm (50 mW·cm⁻²) and 638 nm (100 mW·cm⁻²) laser diodes (Thorlabs). Absorption corrections, data reduction and unit cell refinement were performed using Xia2 and CrysAlisPRO software (Rigaku Oxford Diffraction, 2019)[65]. Introduction of hydrogen atoms in the structures of **2** and **2′** led to the instability of the refinement; therefore, to maintain consistency, we decided to remove hydrogen atoms from the corresponding structural models.

Diffraction experiments for **3** were performed at 100 K (CCDC 2352265) and 293 K (CCDC 2352266) using Bruker D8 Quest Eco diffractometer (Mo Kα radiation, Triumph® monochromator). Absorption corrections, data reduction, and unit cell refinements were performed using SADABS and SAINT programs included in the Apex3 suite. All the structures were solved using direct methods and refined anisotropically using weighted full-matrix least-squares methods applied to $F^2$[66–68].

An ORTEP-style illustration of each structure can be found in the Supplementary Information (Figs. S25-S34). A note with justification on all CheckCif A- and B-level alerts in the reported crystal structures is provided in the Supplementary Information (Note S1).

**Powder X-ray diffraction.** PXRD patterns were collected using Bruker D8 Advance Eco diffractometer equipped with Lynxeye silicon strip detector, Cu sealed tube radiation source and a capillary stage at room temperature. Samples were ground to a powder using an agate mortar inside the glovebox and loaded into glass capillaries 0.3 or 0.5 mm diameter. The capillaries were broken in half inside the glovebox and the open end was sealed using silicon grease before they were moved

to the PXRD instrument and mounted on the goniometer head using bee wax. The simulated PXRD patterns were obtained from the scXRD data using Mercury software[69]. The experimental PXRD pattern for **1** was subjected to background correction using the DIFFRAC algorithm implemented in the DIFFRAC.EVA V5 software.

**Periodic density functional theory (DFT) calculations.** All periodic DFT computations were performed using CRYSTAL23 software[70]. Initial geometrical parameters for low-spin and high-spin state structures (**1** and **2**, respectively) were adopted from the single-crystal X-ray diffraction data (**1** - CCDC no. 2352258; **2** - CCDC no. 2352263). Hydrogen atoms were added according to chemical criteria in the positions of electron density maxima from the diffraction data. Geometry optimizations within Unrestricted Kohn-Sham (UKS) formalism were conducted using several hybrid and range-separated hybrid functionals (details in Supplementary Information) and pob-TZVP basis set[71,72]. Best fit to the experimental data was obtained for HISS middle-range corrected range-separated hybrid functional[73–75]. In each run, a $4 \times 4 \times 4$ $k$-point mesh in the reciprocal space was generated in line with the Monkhorst–Pack methodology[76]. In all computations, tighter tolerances on the exchange and Coulomb integrals were used with the TOLINTEG set to 10, 10, 10, 10, and 20[70]. Numerical integration accuracy was provided using pruned XXLGRID comprising 99 radial and 1454 angular points in the region of chemical interest. The self-consistent field (SCF) convergence criterion was set to $10^{-8}$ a.u. To achieve the correct spin states for the computed systems, the difference between the number of α and β electrons was fixed at all $k$-points during the initial 30 SCF cycles of the first optimization step. The same spin-locking protocol was used to reproduce the spin-state change. The pre-optimized structure of the low-spin state was fully re-optimized with the number of unpaired electrons per unit cell set as expected for two high-spin Mo atoms (α-β = 6) during the initial 50 SCF cycles of the first optimization step. High-spin to low-spin transformation was replicated using an analogous procedure setting the number of unpaired electrons per unit cell as α-β = 2.

**CASSCF/CASPT2 calculations.** Complete active space self-consistent field (CASSCF) and complete active space second-order perturbation theory (CASPT2) calculations of the excitation energies were done with the MOLCAS 8.2 code[77]. **1** was represented as a $[Mo(CN)_7]^{4-}$ unit embedded in 15 $K^+$ ions represented with model potentials and 203 optimized point charges to take into account the long-range Madelung potential. Relativistic corrections were accounted for with DKH Hamiltonian and dynamic electron correlation was calculated on the CASPT2 level. In case of **2**, the $[Mo(CN)_6]^{3-} \cdot CN^-$ moiety was embedded in 16 $K^+$ ions represented with model potentials and 162 optimized point charges that represent the Madelung potential in the $[Mo(CN)_6]^{3-} \cdot CN^-$ region. The molecular orbitals were expanded in the ANO-RCC all electron basis sets: Mo (6s, 5p, 4d, 1f), K (4s, 3p), C and N (3s, 2p, 1d), O (3s, 2p), H (2s)[78]. The active space consists of 9 orbitals (5 Mo-4d, 2 CN-π, and 2 CN-π*) and 7 electrons. The ions used to embed the central unit are represented with the large core Hay and Wadt effective core potentials with a net charge of $+1$[79]. CASPT2 correlates all the electrons except the deep core ones (C, N, O-1s, Mo-1s…3p). The standard IPEA = 0.25 zero$^{th}$-order Hamiltonian was used and an imaginary shift of 0.15 Eh was added to the denominators to avoid the appearance of intruder states. The Cholesky decomposition is used to speed-up the handling of the two-electron integrals with a threshold of $10^{-3}$ Eh.

**X-ray absorption spectroscopy (XAS).** XAS spectra at Mo $L_{2,3}$-edges were obtained at the ID12 beamline (ESRF, The European Synchrotron)[80]. The data were collected using total fluorescence yield detection mode and were subsequently corrected for reabsorption

effects. Compound **2** was obtained in situ by irradiating **1** for 12 h at 4 K, using 405 nm laser with a power of $P = 200$ mW. The X-ray Magnetic Circular Dichroism (XMCD) spectra of **2** were obtained as the difference between two consecutive XAS spectra recorded with opposite photon helicities and corrected for the incomplete circular polarization rate. To ensure the absence of experimental artefacts the measurements were systematically performed for both magnetic field directions. The normalized spectra were analyzed using the magneto-optical sum rules given for $L_2$ and $L_3$ absorption edges[57] to afford effective spin magnetic moment ($M_{S,eff} = -2<S_{eff}> \mu_B$) and orbital magnetic moment ($M_L = -<L_z> \mu_B$):

$$\langle L_z \rangle = \frac{2\langle n_h \rangle}{3} \cdot \frac{I_{L_3}^{XMCD} + I_{L_2}^{XMCD}}{I_{L_3}^{XAS} + I_{L_2}^{XAS}} \quad (1)$$

$$\langle S_{eff} \rangle = \langle S_z \rangle + \frac{7}{2}\langle T_z \rangle = \frac{\langle n_h \rangle}{2} \cdot \frac{I_{L_3}^{XMCD} - 2I_{L_2}^{XMCD}}{I_{L_3}^{XAS} + I_{L_2}^{XAS}} \quad (2)$$

The precise sample temperature and the absolute magnitude of the magnetic moment ($M_{tot} = M_S + M_L$) were determined by scaling the field and temperature dependent XMCD signal intensity to the results of magnetic measurements for **2** (Fig. 3c) following the previously described method[81].

**Infra-red (IR) spectroscopy.** IR spectra were recorded using Nicolet iN10 MX FT-IR microscope in a transmission mode. A polycrystalline sample was spread onto the surface of a $BaF_2$ optical window and sealed under Ar atmosphere inside a Linkam THMS350V temperature-controlled stage. The temperature control was achieved with a flow of liquid nitrogen, and the irradiation experiment was conducted using a 405 nm laser diode with a power $P = 7–8$ mW·cm$^{-2}$.

**UV-visible spectroscopy.** UV-vis spectra for **1** and **3** were measured at room temperature in transmission mode using a Shimadzu UV-3600i Plus spectrophotometer. A solution of **1** was prepared by dissolving a mixture of $K_4[Mo^{III}(CN)_7] \cdot 2H_2O$ (4.7 mg; 0.01 mmol) and crypt-222 (19 mg, 0.05 mmol) in 20 mL of deoxygenated acetonitrile, and then filtered using a syringe equipped with a 0.22 μm pore size PTFE membrane to remove insoluble impurities. Both solutions were placed inside air-tight quartz cuvettes and measured immediately after preparation. Solid sample of **3** was mixed with paraffin oil between two quartz slides and measured with an integrating sphere.

**Electron paramagnetic resonance (EPR).** To enable quantitative in situ irradiation of the EPR sample in the resonator cavity, powder of **1** was deposited as a thin layer on a double-sided adhesive tape glued onto a plastic holder, which was placed in a standard 4 mm X-band EPR tube (Wilmad). Continuous-wave EPR spectra were recorded on a Bruker ELEXSYS E580 spectrometer with a Bruker ER 4118X-MD5 resonator and an Oxford Instruments ER 4118CF helium flow cryostat, at a microwave frequency of 9.628 GHz, with microwave power of 40 μW, modulation amplitude of 0.5 mT and at 10 K.

**Optical reflectivity measurements.** Reflectivity measurements have been performed with a home-built system, operating between 10 and 300 K (at 4 K·min$^{-1}$) and in the range of 400–1000 nm. A halogen-tungsten light source (Leica CLS 150 XD tungsten halogen source adjustable from 0.05 mW·cm$^{-2}$ to 1 W·cm$^{-2}$) was used as the light source for the high-sensitivity Hamamatsu 10083CA spectrometer. The measurements were calibrated with barium sulfate as the reference sample. As the samples are potentially very photosensitive, the light exposure time was minimized during the experiments keeping the samples in the dark except during the measurements when white light is shined on the sample surface ($P = 0.08$ mW·cm$^{-2}$). For all

excitation/de-excitation experiments performed at 10 K, the sample was initially placed at this temperature keeping the sample in the dark to avoid any excitation. Light-emitting diodes (LEDs) operating between 365 and 1050 nm (from Thorlabs) were used for excitation experiments. For the excitation/de-excitation experiments in optical reflectivity, the power and the time of the irradiation were systematically adapted to the optical properties of the material to obtain the fastest possible response and to minimize thermal heating effects.

**Thermogravimetric (TG) analysis.** TG analysis was performed using a NETZSCH TG 209 F1 Libra thermogravimeter under a flow of dry nitrogen gas of 20 mL·min$^{-1}$ and a temperature scanning rate of 2 K·min$^{-1}$.

**Magnetic and photomagnetic measurements.** Magnetic measurements were performed using Quantum Design MPMS3 Evercool SQUID magnetometer in the magnetic fields up to 7 T. Polycrystalline sample of **1** (21.8 mg) was inserted into a polyethylene (PE) bag and sealed under an argon gas atmosphere inside the Inert PureLab HE glovebox ($O_2 < 0.1$ ppm and $H_2O < 0.5$ ppm). The PE bag was attached to the quartz sample holder using Kapton tape. Polycrystalline sample of **3** (32.8 mg) was inserted into Delrin sample holder described elsewhere[82], which was inserted into the plastic straw. The experimental magnetic data were corrected for the diamagnetism of the sample and the sample holder. Photomagnetic measurements were performed for samples (0.3–1.0 mg) placed between two layers of adhesive tape and inserted into the plastic straw. For the relaxation measurements, very small amount of sample (ca. 0.1 mg) was used to facilitate fast and complete conversion from **1** to **2**. The accurate sample mass and diamagnetic corrections were determined by comparison of the data collected in the dark on bulk samples. The irradiation was performed using laser diodes (405 nm – $P = 5$ mW·cm$^{-2}$, 638 nm – $P = 15$-20 mW·cm$^{-2}$).

## Data availability
Source data for all Figures are provided with this paper as a single Excel file and are also available from the corresponding author: Dawid Pinkowicz (dawid.pinkowicz@uj.edu.pl). Crystallographic data reported in this article were deposited with the Cambridge Structural Database and can be obtained free of charge from the Cambridge Crystallographic Data Center (CCDC) via www.ccdc.cam.ac.uk/data_request/cif with the following CCDC numbers: 2352257 (**1**), 2352263 (**2**), 2352259 (**1** relaxed by 638 nm irradiation), 2352264 (**1'**), 2352261 (**2'**), 2352262 (**1'** relaxed by heating to 200 K), 2352258 (**1$_{test}$**), 2352260 (**1$_{test2}$**), 2352265 (**3** at 100 K) and 2352266 (**3** at 293 K). Source data are provided with this paper.

## Code availability
No new code was developed for this study.

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

## Acknowledgements

This work was financed by the European Union within the Horizon Europe Framework Program, ERC Consolidator Grant LUX-INVENTA no. 101045004 (D.P.). Views and opinions expressed are, however, those of the authors only and do not necessarily reflect those of the European Union or the European Research Council Executive Agency. Neither the European Union nor the granting authority can be held responsible for them. The authors would like to thank the European Synchrotron Radiation Facility for beamtime at the ID12 (proposal CH-6581 and CH-6239) and Diamond Light Source for beamtime at I19-2 instrument (proposal MT18908) and the staff of beamline I19 for assistance with crystal testing, data collection and photoirradiation setup. Access to the ESRF was financed by the Polish Ministry of Education and Science—decision no. 2021/WK/11. We also acknowledge the DUS (D.P.) and RSM (M.M.) grants from the Faculty of Chemistry under the Strategic Program Excellence Initiative at the Jagiellonian University. The study was partly carried out using the research infrastructure cofounded by the European Union in the framework of the Smart Growth Operational Program, Measure 4.2; Grant No. POIR.04.02.00-00-D001/20, "ATOMIN 2.0—ATOMic scale science for the INnovative economy". Polish high-performance computing infrastructure PLGrid (HPC Centers: ACK Cyfronet AGH) is acknowledged for providing computer facilities and support within computational grants nos. PLG/2024/017534 and PLG/2023/016609. R.C., C.M., and M.R. thank the University of Bordeaux, the Région Nouvelle Aquitaine, Quantum Matter Bordeaux (QMBx), and the Association Française de Magnétisme Moléculaire for support and funding. D.P., M.M., M.A., L.M., R.C., C.M., and M.R. acknowledge the continuous support of the Center National de la Recherche Scientifique (CNRS) in particular via the International Emergent Actions program in 2023 (Poland/France IEA-302).

## Author contributions

M.M. prepared the reported compounds, performed and analyzed the magnetic and photomagnetic measurements, participated in the IR and XAS/XMCD spectroscopic studies, performed and analyzed scXRD, PXRD, UV-vis and TG experiments, participated in the analysis of all experimental and computational data, acquired funding and wrote the initial version of the manuscript; M.A. participated in the scXRD experiments and refined the crystal structures; L.M. performed the periodic DFT computations, analyzed the results and wrote the relevant part of the manuscript; M.Rams performed and analyzed heat capacity measurements and participated in the analysis of the magnetic data; M.Rouzières built the optical reflectivity setup and performed the optical reflectivity experiments; A.R. performed and analyzed the XAS/XMCD experiments and wrote the relevant part of the manuscript; F.W. and I.O. participated in the XAS/XMCD measurements; T.L. performed and with A.S. analyzed all EPR measurements; C.G. performed and analyzed CASSCF and DFT calculations and wrote the relevant part of the manuscript; C.M. and M. Rouzières performed optical reflectivity measurements and C.M. analyzed them and wrote the relevant part of the manuscript, and participated in the XAS/XMCD experiments; R.C. analyzed the optical reflectivity data, performed and analyzed the XAS/XMCD measurements and revised thoroughly the manuscript; D.P. conceived and supervised the whole project, acquired funding, participated in the preparation of the reported compounds, performed and analyzed the magnetic and photomagnetic measurements, participated in the IR and XAS/XMCD spectroscopic studies, participated in the analysis of all experimental and computational data and wrote the initial version of the manuscript with M.M. All authors contributed to the preparation and revision of the manuscript and accepted its final version.

## Competing interests

The authors declare no competing interests.
