## [Transparent Peer Review file · Nature Communications]

Reversible single crystal photochemistry and spin state switching in a metal-cyanide complex

Corresponding Author: Professor Dawid Pinkowicz

Version 0:

Reviewer comments:

Reviewer #1

(Remarks to the Author)

In this manuscript, Magott et al report a photo-controlled spin-state switching of a Mo(III) metal center involved in a reversible crystal photochemistry. In this single crystal of $K_4[MoIII(CN)_7] \cdot 2H_2O$, the coordination geometry of Mo(III) transitions from 7-coordinate to 6-coordinate as a result of the cleavage of one Mo-CN bond upon irradiation with 405 nm light at low temperatures. This process is reversible by subsequent irradiation with 638 nm light or by heating above 160 K. Corresponding to the bond cleavage and reformation, the spin state of Mo(III) shifts between the low-spin state ($S = 1/2$) and the high-spin state ($S = 3/2$).

The structural transformation was examined with in-situ single-crystal X-ray diffraction, IR spectroscopy, and X-ray absorption spectroscopy (XAS). The magnetic switching was verified through X-ray magnetic circular dichroism (XMCD) and magnetization measurements.

The photo-regulated magnetic switching observed in this simple salt is impressive, and this work warrants publication in a high-impact journal such as Nature Communications.

My suggestions:

(1)Page 1. "Unfortunately, the photoexcited state of SCO systems suffers from rapid thermal relaxation above 80 K, which limits their practical applications as photomagnetic switches at room temperature."

Several SCO compounds have demonstrated high TLIESST > 80 K, in addition to the $[FeII(LN_5)(CN)_2] \cdot MeOH$, The author should rephrase this sentence for accuracy.

(2)I am interested in the thermal stability of this compound, which can be assessed using TG. If the compound remains stable up to 400 K, I encourage the author to measure the magnetic susceptibility above 300 K to investigate the possibility of thermally induced spin-state variations in this compound.

(3)The bond breaking and reformation within the crystal state should be considered lattice effects. I noticed that the free CN anion is in close to the K^+ ion (approximately 3.0 Å). A comprehensive analysis of the overall structure, rather than a focus on discrete molecules, would enhance the understanding of this unique phenomenon. The $Mo(CN)_7$ complex anion are typically employed as bridging ligands to construct multimetallic assemblies. A deeper understanding of these structural effects will advance the design of high-performance photomagnetic materials.

Reviewer #2

(Remarks to the Author)

The investigation of reversible photodissociation-association in a 7-coordinated molybdenum complex under dual wavelength irradiation presents an intriguing advancement in solid-state photochemistry. The study submitted by Magott et al. demonstrates that 405 nm light induces cyanide photodissociation to generate complex 2, while 638 nm light or thermal treatment promotes cyanide re-association to form complex 1. The thorough spectroscopic characterization, particularly the magnetic and photomagnetic studies revealing a permanent $S=3/2$ state of complex 2, provides compelling evidence for this novel phenomenon.

While the experimental work is meticulously conducted and well-presented, several aspects require additional clarification to meet Nature Communications' standards. Specifically, the mechanism and dynamics of the wavelength-dependent

reversible photodissociation-association process need more thorough rationalization. The solid-state specificity of this phenomenon (versus solution) also warrants deeper mechanistic explanation.

Given the potential significance of this work for material development and solid-state photochemistry, I believe this manuscript could be suitable for Nature Communications after major revisions addressing these mechanistic aspects. I recommend the authors provide:

1. A detailed mechanistic explanation of the wavelength-dependent processes
2. Analysis of the solid-state specificity
3. Further investigation of the photodissociation-association dynamics

Upon satisfactory revision addressing these points, I would support publication in Nature Communications.

Major Comments:

1. Second paragraph on Page 2: The authors assert that the oxidation state of Mo remains unchanged during photodissociation. While this argument is plausible and consistent with the XAS data discussed later in the manuscript, the evidence provided is unclear. Specifically, the comparison of bond lengths for Mo–C and C7–N7 does not convincingly support their claim. Furthermore, I would expect that the cyanide bond length in 'free' cyanide (e.g., KCN/NaCN) would differ significantly from that in a transition metal-cyanide complex due to potential π -type interactions, such as those involving Mo–CN. Considering their IR studies, I would also anticipate changes in the C–N bond length. Could the authors provide further clarification on this point?

2. The reversibility of the cyanide group is an intriguing finding that warrants a more thorough explanation. Specifically, how does the dissociated cyanide group rebind to the Mo center under 635 nm irradiation or thermally? What drives this re-association, and what could be the mechanism underlying this photochemical process?

3. Additionally, authors have not fully explained why spin-crossover is suppressed in the high-spin system of complex 2. These are key points that must be included in the manuscript, and authors should address these aspects thoroughly.

Minor Comments:

1. First paragraph on Page 2, sentence beginning "Despite the large displacement of the C7N7...": The term "fully restored" is somewhat ambiguous. I recommend including a brief description of experimental observations, along with quantitative data from the photocrystallographic experiment, to illustrate the structural changes before and after irradiation.

2. First paragraph of the "Solution Photochemistry" section: In this section, the authors discuss the photochemical behavior of the complex in solution. The distinct color difference between the solid-state and solution-state forms of 1 raises the question of whether heptacoordination of cyanide is maintained in the presence of crypt-222 in solution. Could the authors provide additional commentary on this observation?

3. I recommend adding a ChemDraw scheme before Figure 1 showing the photochemical transformation of compound 1. A clear molecular diagram would help readers quickly understand the light-induced structural changes, complementing the crystallographic data.

Reviewer #3

(Remarks to the Author)

The manuscript by Magott et al. reports on the reversible spin state switching by light irradiation in single crystals of the metal cyanide complex $K_4[Mo^{III}(CN)_7] \cdot 2H_2O$. The spin state switching is accompanied by a reversible breaking of one of the Mo–CN bonds, leading to a change of the coordination number at the Mo centre between seven and six. The change in the coordination number is responsible for the high thermal stability of the photoexcited state, up to 150 K. The switching behaviour was investigated using a wide range of appropriate methods such as scXRD before and after irradiation with the different wavelengths, IR and EPR spectroscopy, XAS and XMCD. Furthermore, the switching behaviour was also investigated in solution; however, here, the M–CN bond break was irreversible.

In the introduction, the authors could include the bondage isomerism of nitrosyl complexes, which can be switched reversibly by light in the solid state, not too different from what is discussed here. Furthermore, the authors cite one of their own manuscripts, ref. 34, where they discuss irreversible bond break for an $K_4[Mo^{IV}(CN)_8] \cdot 2H_2O$ compound. However, in the manuscript itself, it is discussed as reversible; bond breaking upon light irradiation and bond formation back to the ground state upon heating above a critical temperature, similar to the system described in the present manuscript. As the two systems are very similar, a comparison of the two compounds would be very interesting and should be included in the manuscript. I am surprised that the authors did not do it right from the start, as important design principles could be revealed. In summary, this is a very nice work done at a very high level with regard to the experimental setup used to clearly characterize the observed phenomenon. However, the reference to already existing very similar systems is missing.

Version 1:

Reviewer comments:

Reviewer #1

(Remarks to the Author)

In the revised manuscript, the authors have added new experiments and data that have resolved my concerns, and I believe the paper now meets the standards for publication.

Reviewer #2

(Remarks to the Author)

In the revised manuscript, the authors have addressed all the reviewers' comments with commendable thoroughness. I greatly appreciate their detailed and thoughtful responses, which clearly reflect a deep engagement with the feedback. The current form of the manuscript has been substantially improved from its already strong original form and now presents an even clearer and more compelling contribution to the field. In my view, no further revisions are necessary, and I recommend the manuscript be accepted for publication in its current form.

Reviewer #3

(Remarks to the Author)

In the revised version of the manuscript, the authors addressed all concerns raised by the reviewers in a very detailed way. Some of the suggested experiments (e.g. time-resolved analysis of the bond dissociation mechanism) were not performed as they would require experiments that would need a much longer time than the time usually allowed for a major revision of a manuscript. This analysis of the authors is correct and I agree that such studies can also be performed in subsequent work.

In summary, all requests were sufficiently addressed by the authors and the revised version of the manuscript is suitable for publication.

The point-by-point response to the reviewers' comments and suggestions

We deeply appreciate the time and effort the reviewers have dedicated to evaluating our manuscript. Below, we provide a point-by-point response to their comments, along with a detailed description of the changes made in the revised version of the paper (all changes are highlighted in yellow in the revised manuscript). We sincerely hope that the reviewers will find the revised manuscript suitable for publication.

Reviewer #1

In this manuscript, Magott et al report a photo-controlled spin-state switching of a Mo(III) metal center involved in a reversible crystal photochemistry. In this single crystal of $K_4[Mo(III)(CN)_7] \cdot 2H_2O$, the coordination geometry of Mo(III) transitions from 7-coordinate to 6-coordinate as a result of the cleavage of one Mo-CN bond upon irradiation with 405 nm light at low temperatures. This process is reversible by subsequent irradiation with 638 nm light or by heating above 160 K. Corresponding to the bond cleavage and reformation, the spin state of Mo(III) shifts between the low-spin state ($S = 1/2$) and the high-spin state ($S = 3/2$). The structural transformation was examined with in-situ single-crystal X-ray diffraction, IR spectroscopy, and X-ray absorption spectroscopy (XAS). The magnetic switching was verified through X-ray magnetic circular dichroism (XMCD) and magnetization measurements. The photo-regulated magnetic switching observed in this simple salt is impressive, and this work warrants publication in a high-impact journal such as Nature Communications.

We greatly appreciate this comment. Investigating this system has been particularly rewarding, given its unique properties and the unprecedented photoswitching behavior observed in the solid state. We have carefully revised the manuscript to further highlight these exciting aspects, and we hope that the reviewer will find the updated version even more compelling.

My suggestions:

(1) Page 1. "Unfortunately, the photoexcited state of SCO systems suffers from rapid thermal relaxation above 80 K, which limits their practical applications as photomagnetic switches at room temperature."

Several SCO compounds have demonstrated high TLIESST > 80 K, in addition to the $[Fe(II)(LN_5)(CN)_2] \cdot MeOH$, The author should rephrase this sentence for accuracy.

We fully agree with this criticism and have therefore rephrased the relevant sentence by adding the word "most":

"Unfortunately, the photoexcited state of most SCO systems suffers from rapid thermal relaxation above 80 K, which limits their practical applications as photomagnetic switches at room temperature."

(2) I am interested in the thermal stability of this compound, which can be assessed using TG. If the compound remains stable up to 400 K, I encourage the author to measure the magnetic susceptibility above 300 K to investigate the possibility of thermally induced spin-state variations in this compound.

In accordance with the reviewer's suggestion, we performed the TG analysis for $\text{K}_4[\text{Mo}^{\text{III}}(\text{CN})_7]\cdot 2\text{H}_2\text{O}$ (**1**) under dry N_2 atmosphere. The experiment shows a significant mass loss beginning already at room temperature (Figure R1). The loss of the two water molecules (7.6 %) is completed above 80 °C (353 K). The TGA plot has been included in the revised version of the paper as Figure S24, and the relevant experimental details have been added in the Methods section.

Figure R1. TGA plot for **1** under dry N_2 flow with heating rate of $2 \text{ K}\cdot\text{min}^{-1}$

Nonetheless, we performed magnetic susceptibility measurements on a tightly sealed sample of **1** up to 350 K. However, no evidence of a thermally induced spin crossover was observed in this temperature range (Figure R2):

Figure R2. Magnetic properties of **1** in the 2-350 K range under applied dc magnetic field of 0.5 T.

(3) The bond breaking and reformation within the crystal state should be considered lattice effects.

We agree that the lattice effects are extremely important in this context. For this reason, we have provided a more detailed description of the overall structure of **1** and **2** (please see below). However, we would like to emphasize that photodissociation is also observed in solution for $[\text{Mo}^{\text{III}}(\text{CN})_7]^{4-}$, which suggests that local electronic effects play the primary role in photoswitching behavior, while lattice effects are responsible for its complete and reversible nature in the solid state.

(3 cont.) I noticed that the free CN anion is in close to the K^+ ion (approximately 3.0 Å). A comprehensive analysis of the overall structure, rather than a focus on discrete molecules, would enhance the understanding of this unique phenomenon.

We agree that cations likely play a role in the photoinduced bond-breaking and reformation processes, and as noted earlier, we acknowledge that our original analysis lacked a comprehensive structural perspective. To address this, we now describe the photoswitching phenomenon through the lens of crystal packing, including an analysis of $\text{C7N7}\cdots\text{K}$ distances and H-bonds before and after 405 nm irradiation. This discussion is supported by a detailed structural diagram visualizing changes in interatomic distances (Figure R3). Both the structural diagram and the accompanying analysis have been incorporated into the revised manuscript and Supporting Information (Figure S3).

We would also like to emphasize that periodic DFT computational analyses of **1** and **2** inherently accounted for all atoms within the unit cell, thereby comprehensively incorporating lattice effects. Furthermore, we have validated this approach by optimizing the crystal lattice geometry for two distinct molybdenum spin states.

(3 cont.) The $\text{Mo}(\text{CN})_7$ complex anion are typically employed as bridging ligands to construct multimetallic assemblies. A deeper understanding of these structural effects will advance the design of high-performance photomagnetic materials.

We fully agree with this statement, as reflected in the final two sentences of the “Conclusions” section of our original submission. However, we believe that even the most exhaustive analysis of interatomic distances in a single compound (such as **1**) cannot fully elucidate their impact on the photoswitching phenomenon or establish robust design principles. Such insights can only be achieved through systematic studies of diverse salts and coordination polymers derived from heptacyanomolybdate(III). Preliminary findings from the assemblies we are currently investigating consistently demonstrate photodissociation behavior. Notably, their relaxation temperatures and reversibility are strongly influenced by structural factors such as dimensionality and interatomic distances.

Figure R3 (Figure S3 in the revised version). Crystal packing diagrams for 1 and 2. Crystal packing for 1 before irradiation is shown in panels A1 and A2 while for 2 after 405 nm irradiation it is presented in panels B1 and B2. The panels on the left show nine neighboring unit cells. The panels on the right visualize in detail the fragments where the cyanide dissociation occurs. A1 and B1 show the dissociation of the Mo-C bonds and the change of CN...K interatomic contacts upon light irradiation (H₂O molecules are omitted for clarity here): after the Mo-C bond photodissociation the C7N7 cyanide is effectively stabilized between two potassium ions with the shortest K...C distances of 2.769 and 2.895 Å. A2 and B2 demonstrate the stabilizing role of the H-bonds which are formed upon 405 nm irradiation between C7N7 and the water molecules designated by the positions of O1 and O2.

Reviewer #2 (Remarks to the Author):

The investigation of reversible photodissociation-association in a 7-coordinated molybdenum complex under dual wavelength irradiation presents an intriguing advancement in solid-state photochemistry. The study submitted by Magott et. al. demonstrates that 405 nm light induces cyanide photodissociation to generate complex 2, while 638 nm light or thermal treatment promotes cyanide re-association to form complex 1. The thorough spectroscopic characterization, particularly the magnetic and photomagnetic studies revealing a permanent $S=3/2$ state of complex 2, provides compelling evidence for this novel phenomenon. While the experimental work is meticulously conducted and well-presented, several aspects require additional clarification to meet Nature Communications' standards.

We thank the reviewer for this very positive feedback. We hope that our revisions fully address the comments and are satisfactory

Specifically, the mechanism and dynamics of the wavelength-dependent reversible photodissociation-association process need more thorough rationalization. The solid-state specificity of this phenomenon (versus solution) also warrants deeper mechanistic explanation. Given the potential significance of this work for material development and solid-state photochemistry, I believe this manuscript could be suitable for Nature Communications after major revisions addressing these mechanistic aspects.

We agree with the reviewer that the photoswitching behavior of heptacyanomolybdate(III) potassium dihydrate in the solid state is highly unique, raising critical questions about its underlying mechanism and dynamics. In this manuscript, we focus on a detailed comparative analysis of the photo-stationary state and ground state using advanced characterization techniques, including synchrotron single-crystal X-ray diffraction, magnetometry, X-ray absorption spectroscopy, X-ray Magnetic Circular Dichroism, IR spectroscopy and electron paramagnetic resonance. To unravel the electronic origins of the wavelength-dependent processes, we performed experimental optical reflectivity studies coupled with high-level CASSCF/CASPT2 calculations. Furthermore, we successfully reproduce the effects of structural transformations via computational methods (periodic DFT). Combined with CASSCF/CASPT2 results, these simulations indicate that the observed changes arise from a spin-state change at the Mo center.

I recommend the authors provide:

1. A detailed mechanistic explanation of the wavelength-dependent processes

As previously mentioned, the spin-polarized periodic DFT computations - detailed in the Supplementary Materials of our original submission - offer mechanistic insights. Enforcing a specific spin state during the geometry optimization drives structural evolution in the crystal lattice:

- a) imposing the high-spin configuration on the low-spin structure of $K_4[Mo^{III}(CN)_7] \cdot 2H_2O$ induces CN^- dissociation

b) conversely, enforcing the low-spin configuration on $K_4[Mo^{III}(CN)_6] \cdot CN \cdot 2H_2O$ promotes CN^- association.

These computational results align remarkably well with the photocrystallographic scXRD experiments (see Figure S17 for overlays of experimental and calculated geometries for **1** and **2**). Furthermore, CASPT2 calculations for the heptacyano- and hexacyanomolybdate(III) suggest that the observed structural changes are initiated by excitations within the spin-forbidden $d-d$ transitions:

a) 405 nm irradiation of $[Mo^{III}(CN)_7]^{4-}$ targets $d-d$ transitions at 373.4 and 460.9 nm which correlated with the cyanide dissociation

b) 638 nm irradiation of $[Mo^{III}(CN)_6]^{3-}$ corresponds to spin-forbidden ${}^4A_{2g} \rightarrow {}^2T_{2g}$ transitions, associated with spin-state changes and potential cyanide association.

This hypothesis, discussed extensively in the “Computational studies” section, is consistent with our current findings. However, a definitive mechanistic explanation will require additional sophisticated measurements, which we plan to pursue following publication of the core results presented herein.

2. Analysis of the solid-state specificity

In our manuscript, we compare the photochemical behavior of the $[Mo^{III}(CN)_7]^{3-}$ anion in two distinct environments, in the solid state compound $K_4[Mo^{III}(CN)_7] \cdot 2H_2O$ and in acetonitrile solution. The crystal packing of the $K_4[Mo^{III}(CN)_7] \cdot 2H_2O$ confines the dissociated cyanide ligand in a very specific position, enabling the reversible re-association upon thermal treatment or red-light irradiation. In stark contrast, irradiation of $[Mo^{III}(CN)_7]^{4-}$ in MeCN solution at room temperature leads to irreversible formation of $[Mo^{III}(CN)_6]^{3-}$, as photodissociated cyanide diffuses away from the Mo center, preventing its re-association. This reversible trapping mechanism represents a key aspect of the solid-state specificity of $K_4[Mo^{III}(CN)_7] \cdot 2H_2O$, as detailed in our manuscript.

In our response to reviewer #1, we included a detailed analysis of the crystal packing of $K_4[Mo^{III}(CN)_7] \cdot 2H_2O$ (before irradiation) and $K_4[Mo^{III}(CN)_6] \cdot CN \cdot 2H_2O$ (after violet light irradiation). This analysis revealed additional electrostatic interactions between the photodissociated cyanide ligand and specific potassium cations, complementing the hydrogen bonds that stabilize the dissociated ligand. We highlighted this stabilizing interplay in our original submission. Periodic DFT calculations fully replicate the experimentally observed behavior of the $K_4[Mo^{III}(CN)_7] \cdot 2H_2O$ lattice, confirming its critical role in enabling reversible cyanide photodissociation. This strong correlation between reversibility and the crystalline environment might suggest limited applicability to other photoswitchable systems. However, two key points counter this assumption:

1- molecular specificity: photodissociation occurs even in solution, indicating it is an intrinsic property of the $[Mo^{III}(CN)_7]^{4-}$ anion itself;

2- generalizability: we have validated photodissociation in multiple derivatives of heptacyanomolybdate(III), including coordination polymers, demonstrating its broad applicability (to be published separately).

Thus, while the $K_4[Mo^{III}(CN)_7] \cdot 2H_2O$ lattice optimizes reversibility, the photoswitching mechanism originates at the molecular level, enabling its extension to diverse architectures.

3. Further investigation of the photodissociation-association dynamics

We believe that elucidating the dynamics of this process would require ultrafast experimental techniques, such as X-ray Free Electron Laser (XFEL) diffraction studies or femtosecond spectroscopy, to probe intermediate states. Such studies would need to be conducted in the solid state at cryogenic temperatures to capture transient species and mechanistic steps. While these advanced methods would provide critical insights into the reversible photodissociation mechanism, they demand significant collaborative efforts with specialized research groups possessing expertise in ultrafast science and access to infrastructure like the European XFEL. We intend to pursue such measurements in future work and plan to publish their outcomes separately. Conducting these experiments within the current revision timeframe is unfeasible, as they exceed both our current technical capabilities and the state-of-the-art in methodology.

Notably, the limited published examples of ultrafast studies in molecular solids (listed below) were reported years after the initial discovery of the target materials, underscoring the huge experimental challenges involved:

- Photoinduced magnetization of a cobalt-iron cyanide PBA reported originally by Sato et al. (*Science*, **1996**, 272, 704-705; DOI: 10.1126/science.272.5262.704) was followed by ultrafast studies 25 years later and published by Cammarata et al. in 2021 (*Nat. Chem.*, **2021**, 13, 10-14; DOI: 10.1038/s41557-020-00597-8)
- Reversible photomagnetism in rubidium manganese hexacyanoferrate Prussian Blue analog reported by Tokoro et al. in 2008 (*Chem. Mater.*, **2008**, 20, 423-428; DOI: 10.1021/cm701873s) was investigated by ultrafast studies 16 years later and reported by Hervé et al. recently (*Nat. Commun.*, **2024**, 15, 267; DOI: 10.1038/s41467-023-44440-3)
- Charge-ordered states in tetraoxolene-bridged iron MOF reported by Chen et al. in 2020 (*Chem. Sci.*, **2020**, 11, 3610-3618; DOI: 10.1039/D0SC00684J) was investigated by ultrafast techniques 3 years later by Banu et al. (*Adv. Opt. Mater.*, **2023**, 12, 2301554; DOI: 10.1002/adom.202301554)

Upon satisfactory revision addressing these points, I would support publication in Nature Communications.

We hope that our responses and the revisions made to the manuscript will meet the reviewer's expectations.

Major Comments:

1. Second paragraph on Page 2: The authors assert that the oxidation state of Mo remains unchanged during photodissociation. While this argument is plausible and consistent with the

XAS data discussed later in the manuscript, the evidence provided is unclear. Specifically, the comparison of bond lengths for Mo–C and C7–N7 does not convincingly support their claim.

We acknowledge the reviewer’s valid point that Mo-C and C7-N7 bond length comparisons alone are insufficient to conclusively determine the oxidation state. However, our assertion that the molybdenum oxidation state remains unchanged (+3) in **1** and **2** is supported by a multidisciplinary experimental framework detailed in the original submission:

- a) XAS spectroscopy: The integrated intensity of Mo *L*-edges for **1** and **2** (proportional to *d* orbital hole count) differs by < 4% (Figure 2b), strongly indicating no change in the Mo oxidation state.
- b) XMCD measurements: The spin magnetic moment of **2** ($M_S = 2.86 \mu_B$) corresponds to three unpaired electrons ($S = 3/2$) localized on molybdenum (Figure 2b). A one-electron redox change would produce an even electron count, inconsistent with this result.
- c) Magnetic susceptibility: the magnetic moment of **2** matches that of **3** (in the 15 – 115 K range Figure 3b), consistent with an isotropic t_{2g}^3 ($S = 3/2$) configuration. Oxidation/reduction would alter the magnetic moment.
- d) IR spectroscopy: Cyanide stretching bands in **2** align with those of **3** and the literature compound $\text{Li}_3[\text{Mo}^{\text{III}}(\text{CN})_6] \cdot 6\text{DMF}$ (Figures 2a and S12), confirming the +3 oxidation state (please see also the discussion of the IR spectra for cyanometallates below).
- e) Structural consistency: Mo-C bond lengths in **2** after photodissociation are identical to those in **3** and $\text{Li}_3[\text{Mo}^{\text{III}}(\text{CN})_6] \cdot 6\text{DMF}$ (Figure S4). We have revised the main text to emphasize this comparison:

*“This process appears to occur with the retention of the +3 oxidation state of molybdenum in **2**, as indicated by nearly identical Mo-C bond lengths to **3** and $\text{Li}_3[\text{Mo}^{\text{III}}(\text{CN})_6] \cdot 6\text{DMF}$ (Fig. S4).”*

In summary, while structural data alone may not provide definitive evidence for the Mo oxidation state due to the flexibility of the Mo’s coordination sphere, the **convergence of XAS, XMCD, magnetometry, IR, and structural comparisons** provides unequivocal evidence that Mo retains its +3 oxidation state in **2**. This multidisciplinary approach mitigates ambiguities inherent to any single technique.

Furthermore, I would expect that the cyanide bond length in 'free' cyanide (e.g., KCN/NaCN) would differ significantly from that in a transition metal-cyanide complex due to potential π -type interactions, such as those involving Mo–CN.

We fully agree with the referee’s comment. As detailed in our original submission (page 2, column 1), the C7N7 bond length of the photodissociated (“free”) cyanide in **2** (1.19(1) Å) differs significantly from the same C7N7, coordinated to the Mo center in **1** (1.16(1) Å) before irradiation. Below is an excerpt from our original manuscript discussing this critical point:

*“The C7-N7 bond length of the photodissociated cyanide in **2** measures 1.19(1) Å, which is typical for a 'free' cyanide, as seen in anhydrous KCN or NaCN.⁴⁶⁻⁴⁸ This value is only*

slightly longer than the average C–N bond length observed for cyanide ligands coordinated to Mo in 1 (1.16(1) Å) or in other cyanometallates.⁴⁹

Notably, the “free” cyanide bond distances of 1.19(1) Å and 1.20(2) Å observed in our two independent experiments for **2** (CCDC 2352263 and CCDC 2352261) closely match literature values: 1.20 Å found from single-crystal neutron diffraction investigation of KCN (ref. 47) and 1.18 Å based on single-crystal neutron diffraction investigation of NaCN (ref. 48). This consistency underscores the reliability of our structural assignments and supports the conclusion that photodissociation generates a “free” cyanide ligand in **2**.

Considering their IR studies, I would also anticipate changes in the C–N bond length. Could the authors provide further clarification on this point?

We would like to clarify that the IR spectroscopy does not directly probe bond lengths but instead provides insights into bond force constants, which are influenced by molecular symmetry and electronic environment. For cyanide ligands coordinated to transition metals, the C-N stretching frequency depends on multiple factors including:

- σ -donation (from cyanide to the metal center),
- π -back-donation (from the metal center to the antibonding π orbitals of the ligand),
- the metal center's electronic configuration,
- symmetry of the coordination sphere,
- crystal packing effects.

For example, solid-state IR spectra of molybdenum cyanometallates with increasing Mo oxidation states: $K_4[Mo^{III}(CN)_7] \cdot 2H_2O$ (**1**), $K_4[Mo^{IV}(CN)_8] \cdot 2H_2O$ and $Cs_3[Mo^V(CN)_8] \cdot 2H_2O$ reveal a pronounced dependence of the C-N stretching frequency on oxidation state (Figure R3). Strikingly, this trend occurs despite negligible changes in average C-N bond lengths:

- 1.159(5) Å for $K_4[Mo^{III}(CN)_7] \cdot 2H_2O$ (CCDC 2352257; compound **1** in this work)
- 1.162(11) Å for $K_4[Mo^{IV}(CN)_8] \cdot 2H_2O$ (CCDC 1914245)
- 1.154(4) Å for $(TBA)_3[Mo^V(CN)_8]$ (CCDC 1117356).

This contrast underscores IR spectroscopy's sensitivity to electronic and environmental factors rather than bond-length variations, making it a powerful tool for probing oxidation states and bonding interactions in cyanometallates.

Figure R3. IR spectra in the cyanide stretching region for $K_4[Mo^{III}(CN)_7] \cdot 2H_2O$ (**1**), $K_4[Mo^{IV}(CN)_8] \cdot 2H_2O$ and $Cs_3[Mo^V(CN)_8] \cdot 2H_2O$ (all prepared and recorded in our laboratory).

Furthermore, metal-cyanide complexes show remarkably consistent C-N bond lengths in the solid state. A search in the Cambridge Structural Database identified 1,060 examples of metal-cyanide complex structures (single-crystal structures with $R_1 < 5\%$, not polymeric) that corroborate this trend:

Figure S4. Histogram of C-N bond lengths (Å) in 1060 homoleptic and heteroleptic metal-cyanide complexes featuring various metal centres, as reported in the Cambridge Structural Database.

As shown by these data, C–N bond lengths in metal-cyanide complexes remain remarkably consistent (1.13–1.17 Å) despite significant variations in metal identity (3d to 5d) and the potential influence of ancillary ligands. This narrow range underscores their limited utility as indicators of metal oxidation state or coordination number.

In this context, the observed shift from 1.159(5) Å in **1** to 1.156(7) Å in **2** was not emphasized in this article, as the difference lies within the experimental uncertainty of the measurements.

2. The reversibility of the cyanide group is an intriguing finding that warrants a more thorough explanation. Specifically, how does the dissociated cyanide group rebind to the Mo center under 635 nm irradiation or thermally? What drives this re-association, and what could be the mechanism underlying this photochemical process?

As discussed above, we believe that direct identification of the intermediate states involved in the 638 nm light-induced reverse process would require ultrafast techniques currently beyond our experimental scope. However, intrigued by this phenomenon during manuscript preparation, we addressed it through a combined experimental-computational approach. Optical reflectivity and computational methods shown that the 638 nm irradiation involves spin forbidden $^4A_{2g} \rightarrow ^2T_{2g}$ transitions in the octahedral $[Mo^{III}(CN)_6]^{3-}$ anion in **2**. These transitions gain intensity due to slight geometric distortions in the Mo coordination sphere, enabling photoinduced cyanide re-association. Please refer to the following paragraph of our original manuscript:

*"[...] the calculated spin-allowed transitions for **2** fall within the deep UV region, while the $^4A_{2g} \rightarrow ^2T_{2g}$ transitions appear at 587.6 and 604.8 nm, matching the de-excitation wavelengths (Table S6 and Fig. S11a). These spin-forbidden transitions gain intensity due to a slight distortion in the octahedral geometry of $[Mo^{III}(CN)_6]^{3-}$ observed in **2**, potentially enabling the light-induced reverse metal-cyanide bond association process."*

Furthermore, we have demonstrated that periodic DFT optimization of the high-spin structure of **2** with one unpaired electron per Mo (mimicking the $^2T_{2g}$ excited state) drives the system back to the structure of **1**, mirroring experimental observations. Please refer to the following text in our original submission:

*"The reverse **2**→**1** transformation was similarly modelled by re-optimizing the structure of **2** for one unpaired electron per Mo. Figs. S14-S16 and Table S7-S8 compare the experimental and optimized geometries of the $[Mo^{III}(CN)_7]^{4-}$ and $[Mo^{III}(CN)_6]^{3-}$ anions in their respective crystal structures."*

The thermally induced relaxation of **2** to **1** follows Arrhenius behavior (Figure 3e–f), confirming the metastable nature of **2**. As temperature increases, thermal population of the high-energy states enables **2** to overcome the energy barrier for the cyanide re-association and relaxation to **1**. This rationale is now explicitly stated in the revised manuscript.

3. Additionally, authors have not fully explained why spin-crossover is suppressed in the high-spin system of complex 2. These are key points that must be included in the manuscript, and authors should address these aspects thoroughly.

Complex **2** adopts an octahedral d^3 electronic configuration, which, by conventional understanding, cannot show spin-crossover (SCO) behavior as SCO is typically restricted to octahedral complexes in d^4 - d^7 electronic configurations. On the other hand, the thermally induced **2**→**1** structural transformation and its activation energy

were thoroughly analyzed in the manuscript. Specifically, we demonstrated that this process follows Arrhenius behavior, with the energy barrier for cyanide re-association determined via temperature-dependent relaxation studies.

Minor Comments:

1. First paragraph on Page 2, sentence beginning "Despite the large displacement of the C7N7...": The term "fully restored" is somewhat ambiguous. I recommend including a brief description of experimental observations, along with quantitative data from the photocrystallographic experiment, to illustrate the structural changes before and after irradiation.

We agree with the referee that the term "fully" restored is ambiguous. Accordingly, we have removed it from the manuscript:

*"Despite the large displacement of the C7N7 cyanide upon photodissociation to form **2**, the original crystal structure is ~~fully~~ restored either through 638 nm irradiation or by heating the crystal to 200 K (Fig. 1c and Table S1-2). Both restored structures closely resemble the original **1** in terms of molecular features: bond lengths, supramolecular arrangement and unit cell parameters."*

The quantitative experimental data were already summarized in Fig. 1, Table S1 and S2. In our view, repeating these results in the main text would be redundant, particularly given the article length constraints set in Nature Communications. Additionally, in compliance with the journal's data availability requirements, the source data for Fig. 1 will be provided as a separate Excel file upon final submission. This approach ensures transparency and accessibility while maintaining clarity and conciseness in the manuscript.

*2. First paragraph of the "Solution Photochemistry" section: in this section, the authors discuss the photochemical behavior of the complex in solution. The distinct color difference between the solid-state and solution-state forms of **1** raises the question of whether heptacoordination of cyanide is maintained in the presence of crypt-222 in solution. Could the authors provide additional commentary on this observation?*

We acknowledge the referee's astute observation regarding the color difference between the green solid state form of **1** and its yellow solution in acetonitrile. However, this apparent discrepancy does not reflect a change in coordination number. Notably, **1** also retains its yellow color in aqueous solution, and it is widely used in this form to prepare coordination polymers containing heptacyanomolybdate(III), confirming the persistence of the 7-coordinate geometry in water. Furthermore, the UV-vis spectrum of **1** in water closely matches that of **1** in MeCN in the presence of crypt-222 (Figure R5), further supporting identical coordination environments in both solvents. Thus, while solvatochromic effects and differences in the coordination geometry may account for the color variation between solid and solution states, the coordination number of Mo(III) remains consistent across phases.

Figure 5. Electronic absorption spectrum of a 0.015 *F* aqueous solution of $K_4Mo(CN)_7 \cdot 2H_2O$.

Figure R5. UV-vis spectrum of 0.015 M aqueous solution of **1** (left; adapted from Rossman et al., ref. 36) compared to the UV-vis spectrum of **1** in MeCN the presence of crypt-222 (right; adapted from Figure S6 of our original submission).

The observed color difference between the solid and solution states of heptacyanomolybdate(III) likely arises from differences in coordination geometry and solid state interactions. In the solid state, the complex adopts a capped trigonal prismatic geometry stabilized by lattice effects such as hydrogen bonding or cations interactions, which influence its electronic transitions and optical properties. In solution, geometry may shift to a pentagonal bipyramidal arrangement, altering its absorption characteristics and resulting in the distinct color change. This structural flexibility is consistent with the computational studies, such as the *ab initio* work by Hendrickz et al. (*J. Am. Chem. Soc.*, **2003**, *125*, 3694-3695), which predicts that the six-coordinate $[Mo^{III}(CN)_6]^{3-}$ exhibits no visible absorption bands. This computational insight aligns with our experimental observation of its colorless nature in solution, further supporting the role of coordination geometry in dictating optical behavior.

3. I recommend adding a ChemDraw scheme before Figure 1 showing the photochemical transformation of compound 1. A clear molecular diagram would help readers quickly understand the light-induced structural changes, complementing the crystallographic data.

We appreciate the referee's suggestion, as molecular diagrams are indeed valuable for visualizing structures of coordination complexes. However, in our specific case, the structure of heptacyanomolybdate(III) is sufficiently straightforward, that we chose to present only the crystal structures of compounds **1** and **2** in Figure 1a-c. We believe this approach effectively highlights our successful elucidation of the crystal structures before and after irradiation using high-quality synchrotron single-crystal X-ray diffraction data, while also maintaining clarity and ease of interpretation for the reader.

Reviewer #3 (Remarks to the Author):

The manuscript by Magott et al. reports on the reversible spin state switching by light irradiation in single crystals of the metal cyanide complex $K_4[Mo^{III}(CN)_7] \cdot 2H_2O$. The spin state switching is accompanied by a reversible breaking of one of the Mo-CN bonds, leading to a change of the coordination number at the Mo centre between seven and six. The change in the coordination number is responsible for the high thermal stability of the photoexcited state, up to 150 K. The switching behaviour was investigated using a wide range of appropriate methods such as scXRD before and after irradiation with the different wavelengths, IR and EPR spectroscopy, XAS and XMCD. Furthermore, the switching behaviour was also investigated in solution; however, here, the M-CN bond break was irreversible.

Thank you for providing such a clear and concise summary of the work presented in our manuscript.

In the introduction, the authors could include the bondage isomerism of nitrosyl complexes, which can be switched reversibly by light in the solid state, not too different from what is discussed here.

We appreciate this valuable suggestion. In response, we have added a brief sentence to the manuscript highlighting bond isomerism in nitrosyl complexes, along with an appropriate reference: Coppens, P., Novozhilova, I., Kovalevsky, A. Photoinduced Linkage Isomers of Transition-Metal Nitrosyl Compounds and Related Complexes. Chemical Reviews 102, 861-884 (2002) (cited as ref. 15 in the revised manuscript):

"[...] or to a lesser extent – inorganic compounds e.g. transition-metal nitrosyl complexes.¹⁵"

Furthermore, the authors cite one of their own manuscripts, ref. 34, where they discuss irreversible bond break for an $K_4[Mo^{IV}(CN)_8] \cdot 2H_2O$ compound. However, in the manuscript itself, it is discussed as reversible; bond breaking upon light irradiation and bond formation back to the ground state upon heating above a critical temperature, similar to the system described in the present manuscript.

Indeed, $K_4[Mo^{IV}(CN)_8] \cdot 2H_2O$ exhibits a somehow similar cyanide photodissociation process; however, in this case, the dissociation is reversible only upon heating above 50 K and cannot be triggered by light. In the sentence: *"While cyanide photodissociation has been extensively studied in aqueous solutions³⁰⁻³³ and, to a lesser extent, in aprotic solvents or solids,^{34, 35} it has always been found to be irreversible."* The words *"or solids"* were inadvertently added during our internal revisions, altering the intended meaning. The original sentence referred exclusively to solutions studies. Thank you for bringing this to our attention. We have now corrected the relevant section to provide a more accurate reference to $K_4[Mo^{IV}(CN)_8] \cdot 2H_2O$ and to clarify the distinction between solution and solid state behavior:

"While cyanide photodissociation has been extensively studied in aqueous solutions³⁰⁻³³ and, to a lesser extent in aprotic solvents, it has always been found to be

irreversible.³⁴ Photodissociation was also observed in the solid state for $K_4[Mo^{IV}(CN)_8] \cdot 2H_2O$,³⁵ but it could only be reversed by thermal treatment.”

(Comment cont.) As the two systems are very similar, a comparison of the two compounds would be very interesting and should be included in the manuscript. I am surprised that the authors did not do it right from the start, as important design principles could be revealed.

The behavior of both systems is indeed similar and merits comprehensive analysis. However, such an in-depth comparison falls outside the scope of the present paper. We plan to address this topic in a separate contribution that will summarize known photomagnetic cyanometallates. Nevertheless, in response to the reviewer’s suggestion, we have revised the manuscript as follows:

- 1) In the sentence “Photodissociation was also observed in the solid state for $K_4[Mo^{IV}(CN)_8] \cdot 2H_2O$, but it could only be reversed by thermal treatment.” we now explicitly note that, in $K_4[Mo^{IV}(CN)_8] \cdot 2H_2O$, the cyanide photodissociation is reversible only by thermal treatment and not by light.
- 2) We have added the following sentence to the Conclusions section: “This contrasts with the previously reported $K_4[Mo^{IV}(CN)_8] \cdot 2H_2O$, where photo-association proceeds exclusively through the thermal route.”
- 3) The final sentence of the Conclusions section has been updated to read:
“By leveraging similar mechanisms in coordination polymers and materials based on the $[Mo^{III}(CN)_7]^{4-}$ building-block, or its analogs such as the aforementioned $[Mo^{IV}(CN)_8]^{4-}$, this approach holds promise for advanced high-temperature photomagnetic applications.”

In summary, this is a very nice work done at a very high level with regard to the experimental setup used to clearly characterize the observed phenomenon. However, the reference to already existing very similar systems is missing.

We greatly appreciate the reviewer’s thoughtful feedback. We trust that the revisions and clarifications regarding our previous work on photodissociation in $K_4[Mo^{IV}(CN)_8] \cdot 2H_2O$ fully address the reviewer’s concerns.

All reviewers (quoted below) recommended publication without further changes. However, the manuscript was revised according to the "Author Checklist" provided by the editorial office.

Reviewer #1 (Remarks to the Author):

In the revised manuscript, the authors have added new experiments and data that have resolved my concerns, and I believe the paper now meets the standards for publication.

Response: Thank you for your support and for the positive recommendation.

Reviewer #2 (Remarks to the Author):

In the revised manuscript, the authors have addressed all the reviewers' comments with commendable thoroughness. I greatly appreciate their detailed and thoughtful responses, which clearly reflect a deep engagement with the feedback. The current form of the manuscript has been substantially improved from its already strong original form and now presents an even clearer and more compelling contribution to the field. In my view, no further revisions are necessary, and I recommend the manuscript be accepted for publication in its current form.

Response: Thank you for your kind words regarding our revisions and for your positive recommendation.

Reviewer #3 (Remarks to the Author):

In the revised version of the manuscript, the authors addressed all concerns raised by the reviewers in a very detailed way. Some of the suggested experiments (e.g. time-resolved analysis of the bond dissociation mechanism) were not performed as they would require experiments that would need a much longer time than the time usually allowed for a major revision of a manuscript. This analysis of the authors is correct and I agree that such studies can also be performed in subsequent work.

In summary, all requests were sufficiently addressed by the authors and the revised version of the manuscript is suitable for publication.

Response: Thank you for your support regarding the subsequent work in time-resolved mechanistic studies and for the positive recommendation.